# A Practical Approach to Providing Environmental Enrichment to Pigs and Broiler Chickens Housed in Intensive Systems

**DOI:** 10.3390/ani13142372

**Published:** 2023-07-21

**Authors:** Temple Grandin

**Affiliations:** Department of Animal Science, Colorado State University, Fort Collins, CO 80523, USA; cheryl.miller@colostate.edu; Tel.: +1-970-443-1510

**Keywords:** welfare, pigs, broiler chickens, environmental enrichment, supply chain

## Abstract

**Simple Summary:**

In many countries, pigs and broiler chickens are housed in intensive systems. Pigs live on concrete floors and straw is not provided. Non-straw enrichments may not be as effective as straw, but they may improve the lives of millions of animals. In many countries, large retail and restaurant buyers have been major drivers of improvements in animal welfare. They used their economic influence to enforce basic welfare standards on farms. They have also been effective at improving animal welfare in slaughter plants. Buyers were often motivated by pressure from animal rights activists to address the issue of animal welfare. On farms, there are two types of welfare programs. They are specialized high welfare niche programs and supply chain programs that maintain basic standards for large commercial farms. This paper will emphasize research literature on practical environmental enrichments for pigs and broiler chickens that will be easy to implement. Pigs prefer enrichment objects that they can chew up and deform. Effective enrichment objects for broiler chickens are peck stones, ramps, platforms, and gradient lighting. It is essential to choose enrichments that the animals will actively engage with.

**Abstract:**

In Europe, regulations contain guidance to maintain high standards of animal welfare. In many parts of the world, large buyers for supermarkets or restaurants are the main enforcers of basic animal welfare standards. They can have considerable influence on improving standards on large commercial farms. Research clearly shows that straw is one of the most effective environmental enrichment for pigs. On some large farms, there are concerns that straw will either clog waste management systems or bring in disease. This paper contains a review of both scientific research and practical experience with enrichment devices that are easy to implement. Pigs prefer enrichment objects that they can chew up and deform. Broiler chickens prefer to climb up on objects, hide under them or peck them. It is always essential to uphold basic welfare standards such as animal cleanliness and low levels of lameness (difficulty walking). It is also important to reduce lesions, and maintain body conditions of breeding animals. An environment enrichment device is never a substitute for poorly managed facilities. It should enhance animal welfare on well-managed farms.

## 1. Introduction

Millions of pigs and broiler chickens in many countries are housed in intensive systems. The pigs are housed on concrete floors and straw is not provided. Straw is one of the most effective environmental enrichments for pigs. This paper reviews research studies on non-straw enrichments that can be easily used in intensive housing. It also contains some information based on the author’s practical experience. The author’s unique perspective is that she has served as a consultant for many different commercial companies. For the last forty-three years, she has visited multiple pork and broiler chicken farms in numerous countries. Her approach to environmental enrichment is to introduce both producers and large corporate buyers to enrichments that can be easily implemented on intensive farms. In this paper, research on environmental enrichment devices that would be easy to use in these operations is reviewed. They may not be ideal, but they could be easily implemented now. Non-straw environmental enrichments are not as effective as straw, but they may improve the lives of millions of animals. In hundreds of existing housing systems in some parts of the world, they may be the only realistic option. Straw will either clog waste management systems or it is not available.

In Europe, EU government regulations contain guidelines for high standards of animal welfare [1]. The author has observed that in many other countries, the major drivers of improving farm animal welfare are large buyers of meat, milk or eggs for supermarkets and restaurants. Pressure from animal rights activists was the initial impetus to motivate buyers to improve animal welfare. After supply chain managers visited the farms and saw poor conditions, they were further motivated to make improvements. The author observed that when buyers made their first trips to farms and observed suffering, the animal welfare issue was no longer abstract. It became a real issue that needed to be immediately addressed. The buyers used their huge buying power to make changes. If a producer wanted to continue to sell to the company, they had to comply with the company’s animal welfare guidelines.

During her work as an animal welfare consultant, the author has visited numerous farms in Europe, United States, Brazil, Uruguay, Chile, Mexico, Central America, China, Australia, and many other countries. International companies, such as McDonald’s Corporation, Sainsburys, Maple Leaf Foods, Tyson Foods, and Costco all have animal welfare programs described on their webpages [2,3,4,5,6]. An animal welfare auditing program for slaughter plants has been very effective [7,8]. She learned from working with many commercial companies that effective implementation required simple guidelines. They had to be easily communicated throughout a worldwide supply chain. Training workshops for the slaughter guidelines were conducted by the author in the United States, Europe, and South America. For each species, the workshop lasted only one to three days. It was conducted both in the slaughter plant and the classroom. Many times the students needed the same simple concepts explained over and over. For example, do you count the number of animals that fell during handling or the number of falls? The author also learned that vague guidelines do not work. Telling people to handle animals properly cannot be effectively implemented. What one person thinks is proper handling, another person may think is abuse.

There are two types of commercial farm animal welfare programs. They are specialized high welfare niche programs, such as Global Animal Partnership [9] and A Greener World [10]. The other types of programs in many countries maintain basic welfare standards on large scale commercial farms. The author has traveled to numerous different countries to train welfare auditors for McDonald’s Corporation. The countries where she trained farm auditors were United States, Canada, Thailand, China, Brazil, Germany, United Kingdom, Australia, New Zealand, and Uruguay. Some of the farms that supply these programs are in parts of the world where commercial buyers brought about major improvements in animal welfare. These farms may be located in countries that either have few laws to protect animal welfare, or the laws are seldom enforced. Some people may be concerned about having lower standards on these farms. In many countries outside of Europe, when buyers enforce basic minimum standards, this improves the welfare of millions of animals. Some examples of key welfare indicators to maintain minimum standards are assessment of space requirements, lameness, hygiene and lesion scoring. Food retailers may have a considerable influence on improving animal welfare [11]. In many supply chain systems, audits are conducted by either the buyers or independent auditing firms [5,6,7,12].

The scope of this paper will be environmental enrichment for large commercial farms in supermarket and restaurant supply chains. The major focus will be on pigs and broiler chickens. Some of the retail programs are now requiring environmental enrichments [2]. This paper will not contain a discussion of the pros and cons of different types of housing. Many corporations are now phasing out of their supply chains, the most restrictive housing systems such as sow gestation stalls and small battery cages for hens [2,5]. Some companies, such as Sainsburys, have completely switched to group housing of sows and cage free for laying hens [3].

## 2. Possible Barriers to Implementation of Environmental Enrichment

The emphasis in this discussion will be on easy to implement practical environmental enrichments for large commercial pork and broiler chicken farms. On broiler farms, the author has observed that producers are more likely to use enrichments that can be easily removed when it is time to catch the birds and transport them for processing. The enrichment devices have to be either picked up and removed by people, or attached to cables so they can be raised up to the ceiling. Raising a device to the ceiling is conducted with the same apparatus that is used to raise feeders and waters before catching. The author has also observed enrichment devices hung on the wall prior to catching. Sturdy lightweight devices were required. The author has observed that pork producers are more likely to use environmental enrichments that do not require a lot of extra work. In the section on pigs, there will be tips based on the author’s experience on ways to provide enrichments that producers may be more willing to use. A review of the literature indicated that possible issues that may be associated with some environmental enrichments are waste management problems, biosecurity concerns, sourcing the materials or possible composting problems [13]. Managers of swine facilities with concrete slatted floors and liquid manure systems are concerned that straw will clog their systems [14]. The amount of straw required to reduce tail biting may cause waste management problems. Animal welfare scientific literature clearly shows that straw is a very effective environmental enrichment for pigs [15,16]. Other concerns about straw is the possibility of bringing in disease. Some welfare programs recommend straw or hay bales for broiler chicken enrichment. DEFRA in the UK recommends that hay or straw bales that are kept outside should be covered to prevent contamination by wild birds [17]. Wild birds can spread avian influenza to domestic poultry [18,19]. In the United States, losses due to avian influenza were very high [20]. Housing systems where broiler chickens are allowed to go outside, may increase their exposure to both wild birds and avian influenza. In the U.S., many producers had to stop allowing outdoor access for broiler chickens. Due to both disease and waste management concerns, this discussion will mainly cover non-straw enrichments.

## 3. Use Enrichments That Animals Will Use

A survey conducted in Germany indicated that the public prefers natural environmental enrichments [21]. Sometimes the most effective enrichments are not natural. For example, dairy cows are highly motivated to use a motorized grooming brush [22]. There are also numerous non-motorized brushes that are now available. One YouTube video showed that cattle immediately started using a non-motorized stationary upright brush after it was installed.

An enrichment device is effective if animals actually use it [23,24]. Motivation can be measured by the amount of work an animal will perform to obtain access to the enrichment [24]. Marian Dawkins at the University of Oxford concludes that the thing an animal values is important [24]. Dawkins suggests that there should be collaborative research with the commercial industry to implement effective enrichment devices [24]. During a welfare audit, auditors should look for evidence that the animals or poultry are actively interacting with the enrichment. Effective enrichments have to be used by animals. The author has observed that broiler chickens will use peck stones (Figure 1). The depressions in the peck stone are clear evidence that the chickens were using it. The ingredients in a peck stone may have an effect on how much the broilers will use it. The industry will often be more willing to adopt an enrichment if it provides a productivity gain. Peck stones are very easy for producers to place in the buildings. They are already being used by one major U.S. company. These enrichments are a manufactured item that are provided by the chicken company. Producers like this because the peck stones require no additional labor. Reports in the industry indicate that some brands of peck stones are preferred by the chickens. Producers should choose peck stones that the birds will actively use.

### 3.1. Effective Enrichments for Pigs

In intensive housing systems, there are some environmental enrichments that may not be possible. Research clearly shows that wallowing in mud is highly rewarding to pigs [25]. This environmental enrichment cannot be used on many intensive systems. The emphasis of this article is on enrichments that can be easily added to hundreds of existing farms. Though not ideal, these enrichments would improve the welfare of millions of pigs. There is also a need to find the most effective enrichment devices that will not cause problems with liquid manure systems. Many producers do not want to use straw [26]. This is due to waste management concerns. There are many enrichment devices available that can be used in these systems [27,28,29]. Pigs prefer things they can chew up and deform [27,28,30]. One of the earliest studies showed, that when given a choice of three suspended objects, rubber hoses and cloth strips were preferred over single chains [27]. The objects were attached to switches that counted the number of pulls (Figure 2). When the objects were first put in the pens, the cloth strips were preferred. However, the rubber hoses maintained sustained interest over a period of seven days. The author learned that some types of hoses broke into pieces and jammed the manure pumps. Another type of hose did not cause this problem. To make implementation of rubber hoses successful, a hose must be chosen that pigs will be motivated to chew, and it will not clog drains or pumps. It is also important to avoid hoses that contain either wire or toxic substances. Today, an automated device is available for measuring pig biting and manipulation of enrichment devices [29]. One of the most effective enrichments was a heavy block of wood placed on the floor [30]. Another object that maintained the pig’s interest consisted of chewable polyurethane balls that were mounted on springs. The entire device was securely fastened to the floor [30].

The Dutch government based the use of single chains [31]. The use of branched chains may be more attractive to pigs than a single chain [31]. The most effective non-straw environment devices were either suspended, bolted to the floor or were very heavy pieces of wood [28]. For finishing pigs, suspended rubber hoses installed in pens with slatted floors provided no increase in weight gain [27]. To avoid problems with waste management, pigs on partially slatted floors can be given up to 20 g of straw per pig each day. Unfortunately, this small amount of straw may not be effective for reducing tail biting [14]. One study showed that the amount of straw could be reduced by providing birch wood pieces [14]. Unfortunately, this material would be difficult and expensive to source in many countries.

Providing nesting material for farrowing sows improved productivity. A meta-analysis of 26 studies showed that lucerne hay or straw nesting materials improved the number of piglets born alive and reduced still births [32]. The sows also had improved maternal behavior and responsiveness to piglet distress calls [32]. For producers who want to avoid bedding, one study showed that hanging a 165 cm × 60 cm burlap cloth in the farrowing stall had some advantages. The sows had a lower percentage of still births and a trend towards more piglets born alive [33]. It is likely that burlap cloth is less effective than hay or straw bedding. This research clearly shows the importance of providing straw for farrowing sows. To encourage producers to perform the extra work that straw will require, they should be shown the results of these research studies. The author has observed that when producers are shown that productivity will be improved, they are more willing to make a change. During her long career, the author has implemented many improvements by showing the financial benefits. A possible compromise, to balance welfare needs against waste management problems would be to provide straw in the farrowing pens and to avoid straw in the gestation and finishing (fattening) pens.

Another possible beneficial effective of environmental enrichment is reduction in the startle response. Finishing pigs raised on concrete slatted floors had a reduced startle response when they had access to suspended rubber hoses [27,34]. In conclusion, in this section about pigs, producers should work to invent new enrichment devices. Producers need to test enrichment devices to find the ones that pigs will continue to use.

### 3.2. Enrichment for Broiler Chickens

This discussion will be limited to standard commercial broiler houses where the birds are raised inside a shed on litter. The author has visited broiler houses in the U.S., Canada, China, New Zealand, Australia, Thailand, Brazil, France, and the U.K. Before the benefits of an enrichment device can be evaluated, the producer must maintain a good litter condition and low ammonia levels. If the litter transfers soil onto the broilers, it is definitely not acceptable. The litter should be friable (easily crumbled) to permit scratching and dust bathing. Broiler producers can easily monitor some signs of poor welfare at the slaughter plant. The major key welfare indicators are hock burn, foot pad lesions, dirty birds, and breast blisters. Providing broilers with elevated platforms and straw bales reduced both hock burn and foot pad lesions [35]. Raised platforms or ramps are an easy to implement environmental enrichment. The chickens will either climb on them or hide under them (Figure 3). There are many different designs of elevated platforms and ramps [36,37,38]. The author has observed broilers hopping up and riding small suspended scales (Figure 4). The birds were actively using it. One advantage of using ramps and platforms is that they reduce the risk of introducing disease with straw bales. Producers will be more willing to implement this ramp because it can be raised to the ceiling along with the feeders and waterer prior to catching.

In broiler houses, the birds often cluster along the walls. Another simple enrichment is adding vertical wall panels in the middle of the house. This greatly increased the broiler’s use of space in the middle of the building [39]. Other areas where it is easy and practical to improve broiler welfare are changes in lighting and reduction in stocking density. There are some lighting conditions that are clearly not acceptable. The author visited an overcrowded broiler farm which had extremely dim lighting. It was impossible to walk through the house without causing piling. The daytime lighting program was so dim that the author could not see the feed and water lines after being in the house for twenty minutes. A recent study showed that reducing stocking density provided the greatest benefit for fast growing broiler chickens [40]. Foot pad lesions were reduced and gait (walking) scores were improved [40]. An easy way to determine if a house is overstocked is to walk through the birds on the day of catch. The birds should be able to move at least one meter away from the person without piling. In the late 1990s, most of the houses in the U.S. had either open sides or white translucent curtains that admitted natural light. When new houses were built, some of them were switched to dark curtains and low light levels. From a welfare standpoint, this may have been a mistake. One study showed that chickens performed more natural behaviors when they had natural light [41].

A practical innovative lighting system has been developed by Karen Christiansen at Tyson Foods. It is called choice lighting or gradient lighting. The birds have bright lighting over the feedlines and dimmer 2–5 lux lighting on the sides [42,43,44]. The lights over the feedlines should be 20 lux minimum and 40 lux maximum. A picture found online showed a series of bright LED lights spaced along the feeder lines [43]. The birds preferred to rest away from the bright light. The birds can choose the lighted area they prefer [44]. The variable “choice” lighting increased both natural behaviors and physical activity [42]. This system would be an easy install in many existing houses.

Another possible lighting innovation is “disco chicken” developed by Anna Johnson and her research group at Iowa State University. Periodically throughout the day, laser beams moved along the floor [45]. The chickens followed and pecked at them. There is some evidence that lasers may improve food intake. This system turns on several times a day. Producers must be careful to not move the laser spots too fast. To gain an idea of how chickens respond to lasers, it is easy to experiment with a small laser pointer that is used for PowerPoint presentations. Veterinarians who have biosecurity concerns about enrichments, such as straw bales, really like lighting enrichments. One veterinarian told the author that the main reason he liked lighting enrichments is that they would never cause a biosecurity issue from avian influenza.

Birds have natural behaviors to (1) sit on elevated areas, (2) hide under things, (3) peck things and (4) scratch and dust bathe. Observations by the author have shown that when broilers become heavy, they stop using perches. The author has observed that when elevated platforms are used, heavy birds will both set on top of them and hide under them. When platforms and ramps are designed correctly, the birds will definitely use them. When chickens are raised outside, they will move further away from the house if they have access to trees and bushes to hide under [46]. This may be an instinctual behavior inherited from their wild ancestors. Chickens quickly eat out in the open and then retreat and hide where they will be safe from aerial predators. This instinctual motivation might explain the effectiveness of “choice lighting”. The birds go hide after eating. To provide a substrate for pecking, the litter should remain loose and friable. Hard, moist litter may reduce natural behavior.

## 4. Music and Animals

Another possible enrichment may be music. The author has observed that playing a radio in a pig barn with a variety of talk and music reduced startle responses when a plane flew overhead or a car drove up. The farm was located near a small airport. For music to be effective, there should be periods where it is off and the animals can have silence [47]. Research with a variety of animals indicates that music should not be over 70 db [46,48]. It is also best to avoid heavy metal or a fast tempo over 100 beats per minute [48,49]. It is recommended to have a variety of music [50] and more mellow music such as classical [51] or Indian music [52].

Does music have any positive effects on production? In dairy cattle, music improved moving into a robotic milking unit [48]. Another study showed that dairy cows exposed to music had a faster milking speed and better milk letdown [51,52]. Music that was played before milking had some beneficial effects [53]. It is important to examine how music is introduced. A sudden introduction of something novel may be stressful [54]. In the dairy cows, there may be a learned response that helps trigger letdown.

In pigs, exposure to classical music during the last third of the pregnancy and during farrowing improved the weight of the pigs [55]. The researcher used classical Bach and there was no repetition of the playlist for two hours [55]. When a QBA (Qualitative Behavioral Assessment) was used, the pigs emotions were judged to be more positive [56]. More research is needed to fully understand the beneficial effects of music on pigs and broiler chickens.

## 5. Environmental Enrichment Does Not Remedy a Poor Environment

Providing enrichment does not compensate for a poor environment due to either overcrowding or poor management. Some examples of good well-managed farms are: carefully maintained equipment that helps prevent injuries, adequate ventilation with low ammonia levels, and gentle quiet handling of the animals. The author has visited poorly managed straw bedded farms which had become mucky because not enough straw was used. The cattle and pigs had become filthy dirty. In another case, young turkeys in the middle of winter looked like they had been dipped in black road tar. Poor ventilation was one of the problems on these farms. In indoor pig farms with partially slotted floors, well-managed systems will have clean pigs that have learned to dung over the slats.

To clearly illustrate the detrimental effects of a poor environment, cattle examples will be used. They clearly show the effects of good and poor management. The first priority in any animal welfare program is to prevent suffering. Poorly designed cow cubicles (freestalls) may cause a high percentage of swollen or injured legs [57,58]. There is a big difference between the best and worse operations for welfare issues such as lameness. Many welfare specialists agree that lameness is one of the most serious welfare issues [59,60,61].

Another area of welfare concern is muddy conditions or a lack of shade for outdoor feedlot cattle [60,61,62]. When cattle breathe with their mouths open, they are heat stressed. Some producers are now using solar panels to shade livestock [63,64,65]. The author has observed that heavy cattle with black hides may be more likely to show signs of heat stress during hot weather. Providing shade in large outdoor feedlots would improve animal welfare [66]. Many cattle feeders are concerned about cost. Using solar panels for shade would solve the cost issue. The electricity generated by the panels would pay for the shade. Over the years, the author has observed that some outdoor feedlots have increased the stocking density in feedlot pens. This may have also contributed to more mud issues. When given a choice, cattle preferred to lay on wood chips compared to a muddy area [67]. They spent more time on the wood chips. The author looked at old photos of cattle feedlots from the 1970s and the pens contained half as many cattle. This may have made problems with muddy conditions worse.

## 6. Possible Future Issues Such as Straw and Labor Shortages

Straw is well documented as an effective environmental enrichment and bedding material for pigs and other livestock [15,16]. To keep the animals in a bedded pack system clean and dry requires large quantities of bedding. A possible threat to straw supplies is cellulosic ethanol. Most ethanol is made from grain, but ethanol systems have been developed to convert fibrous material, such as straw, into ethanol [68,69,70]. If cellulosic ethanol becomes popular, there could be a huge shortage of bedding material. There are two factors that may delay this. Straw is bulky and expensive to ship. The other difficulty is with handling the fibrous material [68,70]. A large cellulosic ethanol plant in Romania has recently started production [69]. There are still technical problems that are slowing the adoption of cellulosic ethanol. If these technical problems are solved, straw could either become more expensive or difficult to obtain in sufficient quantities. If this occurs, grain farmers may have to be contracted by retailers to produce straw for bedding.

Labor may become a greater issue in the future. Environmental enrichment devices are not going to replace good stock people. A possible way to improve the supply of labor is to have stockmanship classes available in high school. Both old and new research studies show the importance of good stockmanship [71,72,73]. Rough handling of dairy cows reduced milk yield [73]. Sows that feared people and moved away had fewer piglets [71]. A recent study showed that stock people who slapped cows and twisted tails had lower milk yield [73]. Higher milk yield was attained when a calm voice and touch was used [73]. Another big future issue is sustainability. This will become a greater issue [74]. The issue of cellulosic ethanol is an example where biofuels that improve sustainability may clash with welfare. Cost is also a big issue. Fortunately, animal welfare improvements will often improve productivity and reduce lesions on the animals [28,32,35,55]. The effects of environmental enrichments on carcass quality are mixed [28]. There is usually little effect. It is the author’s opinion that an increasingly urbanized population will insist on improving the welfare of farm animals.

## 7. Use Pictorial Environmental Enrichment Guidelines

The author has learned from her experience training several hundred animal welfare auditors that using pictures helps to provide clear guidance. Many pictorial guidelines are already available for training people to score the body condition of live-stock, detect lameness or assessment of dirty animals. The author recommends the use of photographs to illustrate examples of environmental enrichments that would comply with a buyer’s welfare guideline and enrichments that would not comply. The peck stone, shown in Figure 1, would be an example of an enrichment that would comply because the birds have actively used it. The pictures should be on either a phone app or plastic laminated cards. Producers, welfare auditors, and buyers can all easily understand pictures.

## 8. Conclusions

Large buyers can use their economic influence to set minimum basic standards for animal welfare for large intensive farms. This would improve the welfare of millions of pigs and broiler chickens on hundreds of existing farms. There are many low cost easy to implement environmental enrichments that can be used. On many intensive farms, enrichments such as straw or mud wallows are not an option. Even though the welfare standard may be lower than some of the farms in Europe, millions of animals would benefit from simple low cost enrichments.

## Figures and Tables

**Figure 1 animals-13-02372-f001:**
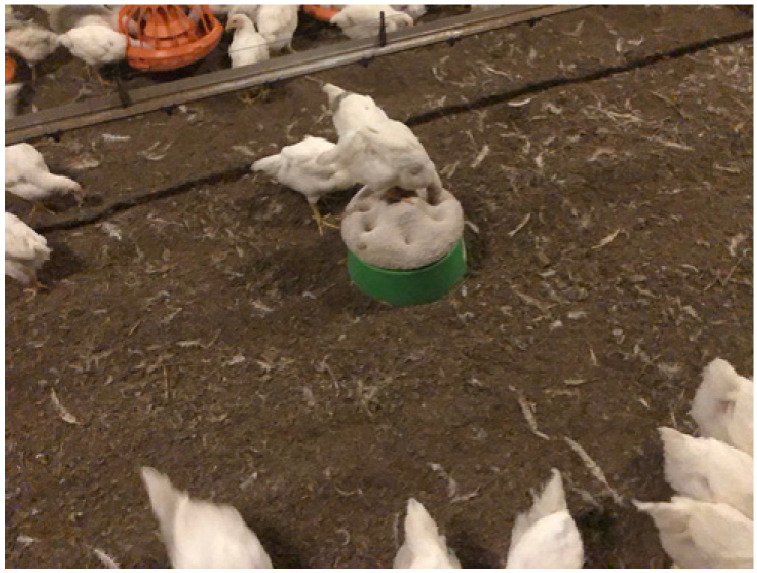
Peck stone for broiler chickens which is being used.

**Figure 2 animals-13-02372-f002:**
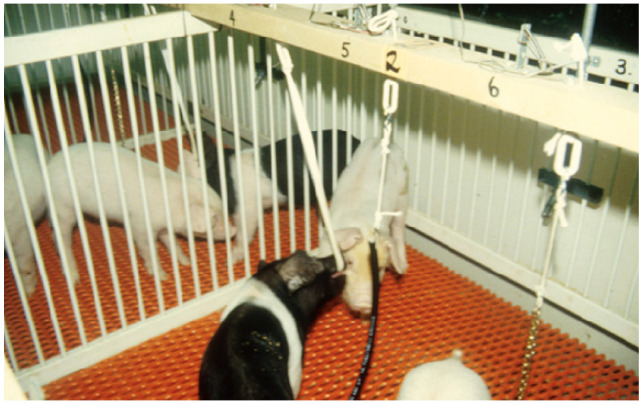
The chain is the least preferred environmental enrichment object.

**Figure 3 animals-13-02372-f003:**
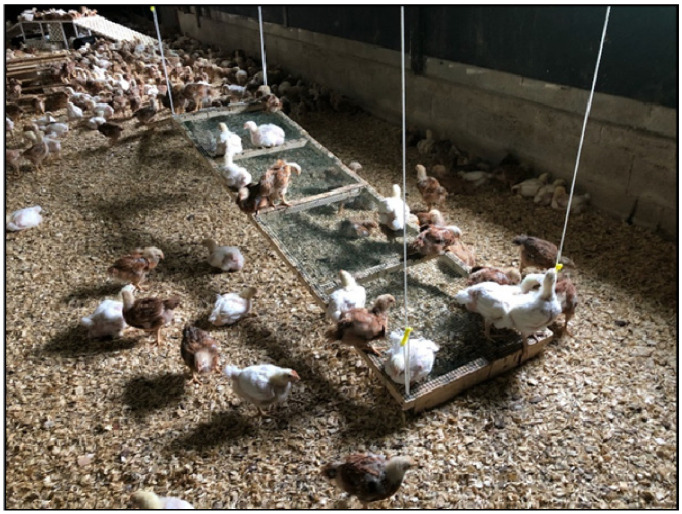
Ramp for chickens to climb on.

**Figure 4 animals-13-02372-f004:**
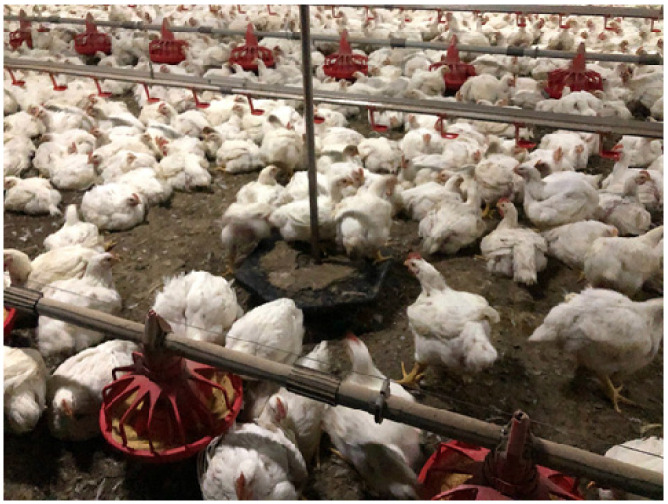
Heavy broilers on the day of catch will ride the scale platform.

## Data Availability

There are no data associated with this paper.

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
