# Peer review of "A Practical Approach to Providing Environmental Enrichment to Pigs and Broiler Chickens Housed in Intensive Systems"

_animals, 2023, doi:10.3390/ani13142372_

Round 1

Reviewer 1 Report

The suggestions / comments:

Lines 2-3: The title of the work shows that the reader should expect comprehensive information on cattle, pigs, and poultry. Meanwhile, information on the practice of enriching the environment for cattle has been presented quite marginally in relation to the other two species. Why did that happen? There is also no information about the environmental enrichment of laying hens - don't large retailers and restaurants deal with this? As far as I know, they often use chicken eggs. This should be clarified and corrected.

Lines 8-9 and 34-36: What about the pressure of animal rights activists, celebrities, and – general – public opinion?

Line 30: where are cattle as the keyword?

Chapter 6. Possible Future Issues Such as Straw and Labor Shortages: these problems have been duly highlighted, but any suggestions to solve them/compensate for straw shortages and/or animal workers would be appreciated in this chapter (or in conclusion).

Author Response

Attached is my revised paper titled A Practical Approach to Providing Environmental Enrichment to Cattle, Pigs, and Poultry to Improve Welfare. To address the concerns of two of the reviewers, I have changed the title to make it very clear that this article is about environmental enrichment for pigs and broiler chickens. The new title is: A Practical Approach to Providing Environmental Enrichment to Pigs and Broiler Chickens Housed in Intensive Systems

Response to Reviewer 1:

Lines 2-3 - I changed the title to make it clear that this article is about pigs and broiler chickens. The new title is: A Practical Approach to Providing Environmental Enrichment to Pigs and Broiler Chickens to Improve Welfare. The laying hens were excluded because I have much more on farm experience with pigs and broiler chickens.
Lines 8-9 and Lines 34-36 – Added additional sentences about the effects of pressure from animal activists.
Line 30 – Since cattle were removed from the title, a cattle keyword is not needed.
Section 6 – Added information on how to address future straw and labor shortages. Possible solutions could be contracting with grain farmers to produce straw and have stockmanship classes in high schools.
The Reviewer’s comments have helped me to greatly improve my manuscript.

Sincerely,

Temple Grandin, Distinguished Professor

Colorado State University

Department of Animal Science

Reviewer 2 Report

A Practical Approach to Providing Environmental Enrichment to Cattle, Pigs, and Poultry to Improve Welfare

Manuscript ID: animals-2378879

Summary

The manuscript discusses the different types of environmental enrichment on largescale poultry and pig farms, as well as alternatives to the use of straw.

General concept comments

Article

It is a very interesting and detailed manuscript that discusses the different types of environmental enrichment in large poultry and pig farms. However, it does not go into sufficient detail about cattle, as it says in the title.

Review

Please proofread the entire text to correct extra spaces between words and sentences.

Sentences within each paragraph are not properly connected to give continuity to the text, please correct.

Specific comments

8-17. Simple summary should be improved to include information on cattle.

18-29. Abstract should be improved to include information on cattle.

59-66. If the major focus is on pigs and broiler chickens, I suggest removing the word "cattle" from title.

Please proofread the entire text to correct extra spaces between words and sentences.

Sentences within each paragraph are not properly connected to give continuity to the text, please correct.

Author Response

Dear Damisa Kaminsin – Thank you for giving me an extension. I just returned from my trip on Saturday.

Attached is my revised paper titled A Practical Approach to Providing Environmental Enrichment to Cattle, Pigs, and Poultry to Improve Welfare. To address the concerns of two of the reviewers, I have changed the title to make it very clear that this article is about environmental enrichment for pigs and broiler chickens. The new title is: A Practical Approach to Providing Environmental Enrichment to Pigs and Broiler Chickens to Improve Welfare.

Response to Reviewer 2:

Thanks for your positive comment about my manuscript. The title of the article was changed to make it clear that the article is about pigs and broiler chickens. Cattle were removed from the title. I proofread the text and corrected the extra spaces.
Line 17-18 – Simple Summary – Information on cattle was not added because cattle were removed from the title.
Lines 19-29 – Abstract – Information on cattle was not added because cattle were removed from the title.
Line 59-66 – Per Reviewer 3’s request, cattle were removed from the title.
The Reviewer’s comments have helped me to greatly improve my manuscript.

Sincerely,

Temple Grandin, Distinguished Professor

Colorado State University

Department of Animal Science

Reviewer 3 Report

Dear Author

Congratulations for this very interesting article and full of learning. I have few suggestions to make.

Line 74- replace the word "research" with "scientific literature"

Line 96 - I found this repetitive phrase "Effective enrichments have to be used by animals"

Line 237 - Please, be more clear on what does well-managed means

Line 253 - Do you consider that muddy conditions in cattle feedlots could reduce the animals' resting time?

Line 289 - Some links are not working. Could you please update them?

All the best!

Author Response

Dear Damisa Kaminsen:

Attached is my revised manuscript Animals-2378879 – A Practical Approach to Providing Environmental Enrichment to Pigs and Broiler Chickens to Improve Welfare. My response to Reviewer 3 are listed below.  The lines numbers for both the original and the revised manuscript are listed.

Response to Reviewer – I appreciate Reviewer 3’s helpful suggestions.

Line 74/82 – The word “research” has been replaced with “scientific literature”

Line 96/106 – To reduce repetition, the words “actively using” were changed to “actively interacting with”

Line 237/252 – Three examples of good well managed farms have been added.

Line 253/275 – Added a reference that shows that cattle spend more time lying on wood chips compared to a muddy area.

Line 325 – The links have been corrected.  I visited the webpages to verify that they still existed.

Temple Grandin,

University Distinguished Professor
Department of Animal Science
Colorado State University
Fort Collins, Colorado 80523-1171
Email: cheryl.miller@colostate.edu

Reviewer 4 Report

I found the article interesting and useful for researchers and producers

Author Response

There were only three reviewers

Reviewer 5 Report

1. Lines 6-7: email of the authors is miss-written “colostate.eldu”: It should be “edu”.

2. In the Introduction section: the contribution of animal welfare to sustainability and productivity should be mentioned. “It is suggested for the purpose of giving an idea only: Dwyer CM. 2020. Can improving animal welfare contribute to sustainability and productivity? BSJ Agri, 3(1): 61-65.”

3. Line 80: Please mention the properties of animal housing systems and their advantages and disadvantages on this issue.

4. Line 103: Under title “3.1. Effective Enrichments for pigs” Wallowing facilities such as mud pools should be emphasized in this issue for pigs.

5. Line 229: Under the title “4. Music and Animals” Please enrich the text focused on why and how music effect because it is a bit uncertain issue.

6. Could you give some information about the effects of enrichments carcass or product quality?

7. In Conclusion section: Please extend the conclusion focused on what is the additional cost of enrichments because broiler and pig industry the profitability is low (profitability on the brink). 

Author Response

Dear Damisa Kaminsin – Thank you for giving me an extension.  I just returned from my trip on Saturday.

Attached is my revised paper titled A Practical Approach to Providing Environmental Enrichment to Cattle, Pigs, and Poultry to Improve Welfare.  To address the concerns of two of the reviewers, I have changed the title to make it very clear that this article is about environmental enrichment for pigs and broiler chickens. The new title is: A Practical Approach to Providing Environmental Enrichment to Pigs and Broiler Chickens to Improve Welfare.

Response to Reviewer 5:

  1. Lines 6-7, corrected edu
  2. Added the Dyer 2020 reference on sustainability at the end of the paper where I discuss the severe animal welfare issues that could occur if cellulosic ethanol depletes supplies of straw. This would be an example of a possible clash between welfare and biofuel sustainability.
  3. Line 80, added a sentence that allowing broilers to go outside was often stopped to prevent exposure to avian influenza.
  4. Line 103, added a reference about the use of mud pools and wallowing for pigs.
  5. Line 229 – Added a sentence that more research will be needed on the beneficial effects of music on pigs and broiler chickens.
  6. Added information and a reference on carcass quality and environmental enrichment.

The Reviewer’s comments have helped me to greatly improve my manuscript.

Sincerely,

Temple Grandin, Distinguished Professor

Colorado State University

Department of Animal Science

Round 2

Reviewer 1 Report

I recommend the manuscript for publication in Animals.